# Tunable, Anisotropic Permeability and Spatial Flow of SLM Manufactured Structures

**DOI:** 10.3390/ma14185205

**Published:** 2021-09-10

**Authors:** Babette Goetzendorfer, Hannah Kirchgaessner, Ralf Hellmann

**Affiliations:** Applied Laser and Photonics Group, University of Applied Sciences Aschaffenburg, Wuerzburger Str. 45, 63743 Aschaffenburg, Germany; s150606@th-ab.de (H.K.); ralf.hellmann@th-ab.de (R.H.)

**Keywords:** additive manufacturing, anisotropic SLM structures, permeability

## Abstract

In this study, we report on a novel approach to produce defined porous selectively laser molten structures with predictable anisotropic permeability. For this purpose, in an initial step, the smallest possible wall proximity distance for selectively laser molten structures is investigated by applying a single line scan strategy. The obtained parameters are adapted to a rectangular and, subsequently, to a more complex honeycomb structure. As variation of the hatch distance directly affects the pore size, and thus the resulting porosity and finally permeability, we, in addition, propose and verify a mathematical correlation between selective laser melting process parameters, porosity, and permeability. Moreover, a triangular based anisotropic single line selectively laser molten structure is introduced, which offers the possibility of controlling the three-dimensional flow ratio of passing fluids. Basically, one spatial direction exhibits unhindered flow, whereas the second nearly completely prohibits any passage of the fluid. The amount to which the remaining orientation accounts for is controlled by spreading the basic triangular structure by variation of the included angle. As acute angles yield low passage ratios of 0.25 relative to continuous flow, more obtuse angles show increased ratios up to equal bidirectional flow. Hence, this novel procedure permits (for the first time) fabrication of selective laser molten structures with adjustable permeable properties independent of the applied process parameters.

## 1. Introduction

Since additive manufacturing (AM) has emerged to an established production technology, in turn stimulating innovative engineering processes, selective laser melting (SLM), as a specific example, provides an opportunity to easily fabricate novel metal structures. The potential freedom of design allows for innovative geometrical approaches. The aspect of reducing weight by use of lattice structures or unsorted foam-like porous materials in contrast to bulk metals especially encourages intense research in many fields of application such as the medical, automotive, or aeronautical engineering field [1,2,3,4,5,6].

Geometrically defined lattice structures yield lightweight applications and are therefore often studied regarding mechanical stability [7,8,9,10,11]. By variation of the fundamental cell geometry, a wide range of possible lattice structures is accessible [12,13]. Moreover, numerous diverse materials are available, as for example in the studies of Leary and Mazur et al. [14,15,16] in which different lattice structures of AlSi12Mg, Ti6Al4V, or Inconel have been are fabricated and characterized. 

Typically, a geometrically periodic unit cell is the basis of a CAD (computer-aided design) designed lattice structure. In SLM, the laser spot diameter, as being typically in the range of 100 µm, determines a fundamental and minimum boundary condition for the strut size of these structures [17]. In addition, the solidification process in SLM is highly sensitive to the applied process parameters, so that careful coordination of building parameters is mandatory in order to obtain rugged and precise structures. For small sized lattices (below 1 mm strut size), a common error source within the powder-based process is redundant powder adhesion on the structure walls [18,19], leading to irregular inner pore geometries, whereas varying process conditions may lead to stochastically distributed partial molten features. Yet, the latter may also be used for obtaining stochastically porous metals utilizing insufficient cohesion and therefore areas with removable powder fractions, resulting in intermaterial pores and channels [20,21,22].

As AM pays off for fabrication of highly individualized parts, such SLM fabricated porous metals have been used for medical implants, benefitting from an improved cell adhesion on porous structures and the ability to mimic the mechanical load conditions of bones [23,24,25,26]. Besides the almost classical Titanium alloy Ti-6Al-4V [27,28,29], various materials have been investigated, such as porous Ni/Ti alloys [30], Tantalum [31], or even porous Titanium with immobilized silver nanoparticles to mortify bacteria and prevent biofilms [32]. Beyond medical implants, porous metal structures are used in fields of catalytic research [33] or as a filtering medium [13,34,35]. Although the stochastic nature of porous material is associated with a random distribution of the pore geometry, reproducible and predictable results in pore size distribution are recommended and therefore controlled process parameters are preferable. For instance, the influence of hatch distance, scan speed, and laser power has been studied in relation to the resulting porous structure of SLM parts [36]. In particular, variation of the hatch distance during build-up even allows for producing functionally graded materials [37].

In this contribution, this known correlation between process parameters and resulting porosity is utilized to further develop SLM structures of defined permeability. Although permeability is directly linked to porosity, sole knowledge of porosity is not sufficient to predict permeability properly since the tortuosity (the way the pores wind through the bulk material) also affects the permeability [38].

Analogue to investigations of gas permeability of porous foam metals [39,40], we assume the same relation between pore geometry and permeability for regular structures. However, in contrast to the mentioned studies, we directly link the pore size to underlying printing parameters by fabrication of defined single line geometries. Thus, variation of printing parameters influences porosity in a controlled way and therefore offers an opportunity to produce materials with tunable permeability.

Finally, the methodology is adapted to produce geometric structures, which exhibit anisotropic permeability properties due to their basic structure. Anisotropic permeability is investigated for stochastically porous materials with regard to gaseous ambiance [41] but so far controlled fluid passage independent of flow direction relative to cell geometry has not been reported. Essentially, the flow control is based on a triangular shaped structure, which (independently of spatial orientation) allows unhindered flow, blocked fluid passage, and a third variable direction. The amount to which a fluid can pass in this specific direction is correlated with the opening angle of the basic structure. Thus, the deliberate design of structures in accordance with controlled process parameters yields in spatial flow ratio control of single line SLM structures.

This new approach combines two aspects. On the one hand, single line geometries represent the smallest possible structures which can be fabricated by SLM. Besides their mechanical suitability for lightweight applications, their spatial fluidic properties become interesting in this dimensional scale. On the other hand, the permeability characteristics of these minimized structures are correlated with the underlying geometric shape, as the geometric degree of freedom in AM techniques offers the opportunity to produce innovative structures. In contrast to stochastically porous media, our study introduces the idea of allowing or prohibiting specific flow directions which can be used in terms of filtering, mixing of fluids, or flushing without removing the filtering part.

## 2. Materials and Methods

### 2.1. Porosity and Permeability

Usually, porous materials are characterized by their porosity, which describes the fluid storage capacity of a material, and by their permeability. Different kinds of pores contribute to the porosity of a material, as they can appear interconnected, isolated, or a non-continuous type [42]. In turn, the permeability of a material is determined by the amount of its continuous pores, as they enable connected channels. Blind (non-continuous) and closed (to the surface) pores do not contribute [43]. Therefore, permeability presents the actual ability of porous media to allow the passage of fluids [44], whereas porosity describes a static fluid storage parameter.

In our study, we produce geometrically defined channel pore structures by adjusting the printing parameters. Basically, the underlying printing parameters are chosen to result in non-porous bulk material. Then, cavities are introduced by stretching the hatch distance between two adjacent sintering lines. Thus, geometrical defined channel structures are fabricated which can be used to estimate the possible fluid passage area. In order to derive advantage from this particular geometric control and knowledge of the applied process, the channel porosity can be calculated with respect to the printing parameters and the channel pore geometry. Thus, a mathematical correlation of the channel porosity and the permeability including printing parameters is possible.

The porosity ɸ is defined in general as the ratio of the pore volume to the bulk volume [45]; in the case of periodic channel structures this porosity can be calculated two-dimensionally by the channel cross sectional area divided by the overall lateral area. As we fabricate geometrically defined cavities, which result in periodic channels with solid surrounding walls, we adapt this definition to our structure. Porosity therefore describes in our case the relation between cavity area to overall surface and the channels are described as pores with a defined geometry. In this case, the Kozeny-Carman equation [46] describes the correlation between porosity ɸ and circular pore geometry (Equation (1)), which can be applied for square patterned and hexagonal pore shapes [47]:(1)$$ɸ=\frac{V_{p}}{V_{b}}=\frac{nA_{p}L_{p}}{A_{b}L_{b}}=\frac{n\pi r^{2}\tau }{A_{b}}\;,$$
with *V_p_* being the pore and *V_b_* the bulk volume, *n* the number of pores, *A_p_* being the cross sectional area of flow tubes, *A_b_* the cross-sectional area of the bulk, *L_p_* and *L_b_* being the pore and the bulk length, *r* the pore radius, and *τ* being the tortuosity *= L_p_/L_b_*. 

The tortuosity τ describes the extent to which a channel is warped relatively to a straight passage and is therefore linked with pore geometry [48]. In our study, the printed channel pores are designed to pass unidirectional through the bulk so that τ can be assumed as 1. The porosity therefore can be calculated according to Equation (1) by the ratio of the pore area divided by the overall area. 

In this context, the permeability of a porous material specifies the possibility for a liquid medium to pass through the material. The volumetric flow rate *q* per area *A_b_* is described by Darcy’s law [49]
(2)$$\frac{q}{A_{b}}=v_{f}=-k_{f}\cdot i,$$
as a filter velocity $v_{f}$ containing the permeability coefficient *k_f_* (with the dimension of a fictive filter velocity) and the dimensionless hydraulic gradient *i.* The minus sign illustrates the downward fluid direction. As the hydraulic gradient i can be written as the ratio of the height difference ΔH divided by the specimens length *L_b_* [50]
(3)$$i=\frac{\Delta H}{L_{b}},$$
*k_f_* equals [51,52]
(4)$$-k_{f}=\frac{q\cdot L_{b}}{A_{b}\cdot \Delta H}=K\cdot \;\frac{\rho \cdot g}{\mu },$$
with permeability *K,* the viscosity of the fluid *µ*, gravity acceleration *g* and density of the fluid *ρ*.

As *ρ**∙g**∙*Δ*H* describes the hydrostatic pressure difference Δ*p*, the flow rate *q* can also be written as [46]
(5)$$q=\frac{KA_{b}}{\mu }\frac{\Delta p}{L_{b}}=n\frac{\pi r^{4}}{8\mu }\frac{\Delta p}{L_{p}},$$
regarding hydrodynamic behavior according to Hagen Poisseuille. 

To obtain a correlation between porosity ɸ and permeability K regarding underlying printing parameters, we combine Equations (1) and (5) in our study to:(6)$$K=\frac{n\pi r^{4}}{8A_{b}\tau }=\frac{r^{2}ɸ}{8\tau^{2}},$$

The coefficient *K* has dimensions m^2^ or *D = Darcy* = 10^−12^ m^2^ and is called the specific permeability or intrinsic permeability of the medium [53]. In the case of single-phase flow, the expression is simply abbreviated to permeability. It appears to be independent of the nature of the fluid, only conditioned by the geometry of the structure. As a result of Equation (6), for defined pore geometries and porosity the permeability *K* should be predictable and moreover adjustable.

### 2.2. Experimental Setup for Permeability Measurement

For investigating the permeability of different SLM structures, a custom-built setup according to ISO 17892-11 [54] is applied as illustrated in Figure 1.

Defined hydrodynamic conditions are assured by two aspects. First, a Mariotte’s bottle ensures uniform fluid pressure and therefore steady flow velocity. As long as the siphon is always at the same position (Height 1) and underneath the liquid level, water will flow out in relation to atmospheric pressure, regardless of the changing water level within the bottle.

Second, the height difference between the siphon of the Mariotte´s bottle and the reading point (Overflow/Height 2 in Figure 1) is kept constant for all probes. In addition, experiments are conducted under controlled temperature conditions using deionized water.

SLM build specimens are applied using a custom-built sample holder. The flow velocity is investigated by measuring the time needed for a defined fluid volume to pass through the SLM structures. By analyzing the flow rate per defined sectional area according to Equation (2) under these controlled experimental conditions, the intrinsic permeability *K* (Equation (4)) is obtained.

### 2.3. Selective Laser Melting

Selective Laser Melting is performed using an EOS M290 (EOS GmbH, Krailing, Germany) equipped with a 400 W single mode fiber laser (YLR-400-WC, IPG Laser GmbH, Burbach, Germany) with a focus size of 100 µm and a laser beam quality of M^2^ = 1.1. Detailed information about the underlying SLM process including technical data is described by Nakano [55]. All prints are performed in nitrogen atmosphere using PH1 material, corresponding to stainless steel 1.4540. The applied energy density of a printed structure can be calculated by Equation (7) [56]
(7)$$E_{v}=\frac{P}{v\cdot h\cdot l},$$
with *E_v_* being the volume energy density, *P* the laser power, v the scan velocity, *h* the hatch distance, and *l* being the layer thickness, which is kept constant at 20 µm for all substrates. Every SLM pattern is fabricated threefold, and every permeability measurement is repeated three times for each single sample for statistical evaluation. Contrary to the typical fabrication of dense substrates, this study deals with the formation of single wall geometries. Therefore, an initial parameter study is required including the described parameters laser power, scan velocity and hatch distance. The technical setup (focus position, build plate position, powder quality) was maintained unmodified in all experiments.

Optical characterization is performed using a Leica DM 6000 microscope (Leica, Wetzlar, Germany) by analysis of three different pore areas per sample and averaging over similar sample sets.

## 3. Results and Discussion

### 3.1. Single Walls

In order to maximize the design freedom for generating porous structures, initially, the formation of adjacent single walls is investigated with respect to wall thickness and minimal feasible wall distance as a function of hatch distance during SLM. Figure 2 shows the formation of single walls in dependence of increasing hatch distances.

Typically, the hatch distance in SLM is chosen to guarantee melting and joining pathways in order to ensure dense printed parts with high mechanical strength and low surface roughness. For generating porous structures, however, the hatch distance has to be selected considerably wider than the laser diameter (100 µm), leading to separated single walls. To ensure sufficient stability of thus generated single walls, the laser energy density must be adapted carefully to warrant wall stability while at the same time impediments like warping or adhesion of surrounding metal powder are to be avoided [57]. Figure 2a shows variations of the hatch distance from 0.15 mm to 0.225 mm with constant laser power (210 W) and scanning speed (1390 mm/s), corresponding to an energy density between 1.01 and 0.67 J/mm^2^, respectively. Using a small hatch distance leads to a relatively dense object (left up), but with increasing hatch distance single lines occur. In Figure 2b, the hatch distance is varied from 0.25 mm up to 0.55 mm and single line walls are fabricated.

Based on these results, a set of parameter variation is defined to cover suitable hatch distances, scanning speeds and laser power values. If the scanning speed is too low, warping due to a high energy density occurs, resulting in build crashes and print abortion. On the other hand, if the scanning is too fast, the energy density is insufficient leading to instable line formation and tilted walls (Figure 3b).

As a result, an appropriate scan speed of 1390 mm/s produces stable single separated wall structures for nearly all laser power and hatch distances (0.25 mm to 0.55 mm). Figure 3a highlights single separated walls using the best parameter set of scan velocity 1390 mm/s and 210 W laser power, corresponding to 0.60–0.27 J/mm^2^). It is noteworthy, that the energy density on its own does not provide enough parameter information for fabrication of single walls as the knowledge of the applied hatch distance is crucial and mandatory for evaluation.

Figure 3a also shows that the width of the single walls lies in the range between 120 and 140 µm which is in good agreement to previous single line studies [17,58]. This basic investigation is now adapted to fabricate two-dimensional single line geometries assuming that variation of the hatch distances yields different pore sizes presupposing different permeable properties. 

### 3.2. Square Pattern

With the knowledge of adequate printing parameters leading to stable single walls, different patterns are constructed. First, a simple square structure is investigated (Figure 4) by combining two orthogonal lines (blue lines in Figure 4a). The distance between two adjacent lines (red arrows) correspond to the chosen hatch distance. As the built SLM walls possess a certain width, the pore size should be in the range of the hatch distance minus wall width (Figure 5).

As the applied laser energy is enhanced by double scanned areas at the crossing points, the basic laser power must be reduced to 175 W to assure stable process conditions.

Variation of the hatch distance (i.e., the distance between two adjacent lines, red arrows (Figure 4a) from 0.3 mm to 0.36 mm results in rising pore sizes (180 to 250 µm) maintaining a steady web width of about 115 µm. Figure 5 depicts the rising pore diameter by unchanged width strip.

On the basis of these results, the porosity ɸ is calculated for the investigated structures using the experimental geometric values for the pore area divided by the overall area:(8)$$ɸ=\frac{{\left(hatch-width\;strip\right)}^{2}}{hatch^{2}}=\frac{pore\;diameter^{2}}{{\left(pore\;diameter+width\;strip\right)}^{2}},$$

The permeability was analyzed using the experimental setup described in Figure 1 and calculated in Darcy [D] = 10^−12^ m^2^. A permeability of 1 D means that by passing this medium under standard conditions, water scores a flow rate of 1 cm per second.

The correlation between measured permeability and printing parameter based porosity is shown in Figure 6, exhibiting a linear relation indicating that the permeability can be fabricated in a controlled way. An increase of 20% in porosity leads to a distinct rise in permeability from 20 D to nearly 100 D. In comparison to literature, similar regular SLM structures exhibit permeability values in the same range [59], confirming our experimental and mathematical framework. Stochastically porous materials with similar pore sizes and porosity ranging from 0.39 to 0.67 scores permeability values up to 16 Darcy, too [60].

Although the linear behavior strengthens our approach to produce tunable permeability materials, it should be kept in mind, that the SLM pore geometries do not possess perfect channel structures. The internal surface is rather rough and insufficiently molten areas could exhibit small porous defects, which disturb the passage of the fluid and influence fluid flow. Therefore, precise adjustment of the laser parameters is recommended to minimize fluctuation in permeability.

### 3.3. Honeycomb Pattern

Next, the single line strategy is extended to a more complex honeycomb structure (Figure 7). Therefore, in contrast to the square pattern, no continuous laser scanning takes place but many single segments in random order are laser-drawn separately.

In case of this geometry, the hatch distance describes the interval between two parallel hexagon side bars (red arrow in Figure 7a). Due to this discontinuous scanning procedure, the resulting structures are less homogenous than the square pattern, but nevertheless a clear correlation between hatch distance and corresponding pore geometry can be stated as shown in Figure 8.

Analogue to the square pattern, the porosity of the honeycomb structure is calculated by use of the experimental geometrical data according to:(9)$$ɸ=\frac{\frac{\sqrt{3}}{2}\cdot {\left(hatch-width\;strip\right)}^{2}}{\frac{\sqrt{3}}{2}\cdot hatch^{2}}=\frac{pore\;diameter^{2}}{{\left(pore\;diameter+width\;strip\right)}^{2}}\;,$$

Figure 9 shows the correlation of porosity versus permeability of honeycomb shaped pores. In contrast to the square patterned structure, a higher deviation emerges although the linear relationship is still recognizable and the permeability values are in a reasonable dimension.

The observable, as compared to the results shown in Figure 8, larger variations from a linear behavior can be assigned to the random scan sequence. If adjacent segments are fabricated successively, a locally higher energy density is achieved compared to more distant scan orders. Therefore, it is more likely to yield non perfect pore geometries due to powder adhesion on “higher energy” pores where surrounding powder particles adhere to the lateral surface [61]. As a result, the complex honeycomb structures are not as regular as the simpler square pattern, illustrating the limits of single line SLM samples. Structural defects such as impermeable pores or constricted channels can lead to deviations in permeability [62]. To avoid this irregularity, non-closed single line structures should be taken as basic geometries, minimizing local energy clusters during processing. Nonetheless, a linear increase with rising porosity can be stated for the honeycomb structure as well. As this particular structure usually is fabricated by common CAD design, the proposed single line process defines the minimum achievable pore size of SLM honeycombs. Despite the variance in permeability, the basic procedure is yet again successfully demonstrated for this geometry.

### 3.4. Anisotropic Structure

The previous findings form the basis for the consequent development of a single line SLM structure with anisotropic permeability. As the basic structure should be line shaped due to energy density control and variable in shape at the same time, a triangularly tilted line structure is constructed. (Figure 10). The tilt angle α is varied from 60° to 150°, conceivable as an opening of the bars from being folded to stretched (see Figure 10a).

As illustrated in Figure 11, the permeability is therefore related to the flow direction in three-dimensions. Direction 1 (Figure 11, blue) allows relatively unhindered flow, as the tortuosity equals 1 due to straightforward channels. On the contrary, the convoluted triangles in direction 3 (Figure 11, orange) prohibit fluid passage nearly completely and act as labyrinthine walls. Interestingly, the permeability in direction 2 (Figure 11, green) corresponds to the opening angle of the triangular structure. The more obtuse α, the more the triangular shape is stretched, the lower the tortuosity resulting in higher flow velocity.

Figure 12 shows the permeability of different enclosed angles α for different flow directions as described in Figure 11. The hindered orientation 3 (orange) acts as assumed and shows low permeability of 10 D or even less with stretching triangle bars. Direction 1 (blue) possesses a permeability more than one higher order in magnitude higher for all angles, indicating unhindered fluid passage with permeability values around 100 D. Orientation 2 (green), reveals a clear correlation of permeability and enclosed angle. For an acute angle of 60° the fluid passage is slower than for more obtuse angles. Yet, an angle of 150° results in equal permeability to direction 1.

To facilitate comparability, Figure 13 contains the relative permeability in respect of direction 1, which is arbitrarily set as 100 D. In dependence of the angle, the ratio of the fluid flow in direction 2 is adjustable. Acute angles lead to a rate of 1:4, whereas a right angle allows a rate of 1:2. Spreading the bars degrades the difference in ratio up to an equal flow for 150°.

Overall, this means that it is possible to control the passage of a fluid in a spatial manner. The tilt angle of the underlying SLM single line structure is responsible for the ratio of fluid passage in direction 2.

## 4. Conclusions

In our study we describe a new approach to fabricate SLM structures with controlled permeability. Basically, the formation of single walls is investigated and the obtained printing parameters are then carefully adapted to create more complex stable single line structures. Initially, a square pattern is successfully realized by combination of two orthogonal selectively laser molten lines. Subsequently, a honeycomb structure is fabricated by a suitable single line scan strategy. As the final aim is to predict the resulting permeability, the crucial point is to control the pore size and therefore the porosity of the structure by applying defined building parameters. As the single line strategy leads to constant width strips, the pore size is tunable and, therefore, porosity, too.

A custom-built experimental set up is used to measure the permeability of the printed structures and a mathematical correlation between printing parameters, especially hatch distance, and resulting permeability is postulated. Both square pattern as well as honeycomb structures exhibit linear relation of porosity and permeability, confirming our mathematical approach. Therefore, SLM metal substrates can now be fabricated with controlled permeability characteristics by adjusting cell structure and printing parameters.

Furthermore, the identified procedure is taken as a basis for the development of an anisotropic single line structure. A triangular shaped line structure is fabricated by the established single line process and the resulting permeability is investigated with respect of spatial orientation. One direction allows nearly unhindered passage of a fluid, whereas a second direction blocks flow-through for the most part. Interestingly, the remaining third spatial orientation is adjustable concerning the tilt angle. If the enclosed angle is acute, the flow ratio relative to the unhindered flow is in the range of 0.25. However, the more obtuse the triangular angle, the more fluid can pass in this specific orientation. Therefore, the ratio rises up to equal permeability to unhindered flow.

In conclusion, through careful design of the basic single line structure and by knowledge of the process parameters, tunable anisotropic permeability, and spatial flow ratio, control of defined single line selective laser melting structures is possible.

## Figures and Tables

**Figure 1 materials-14-05205-f001:**
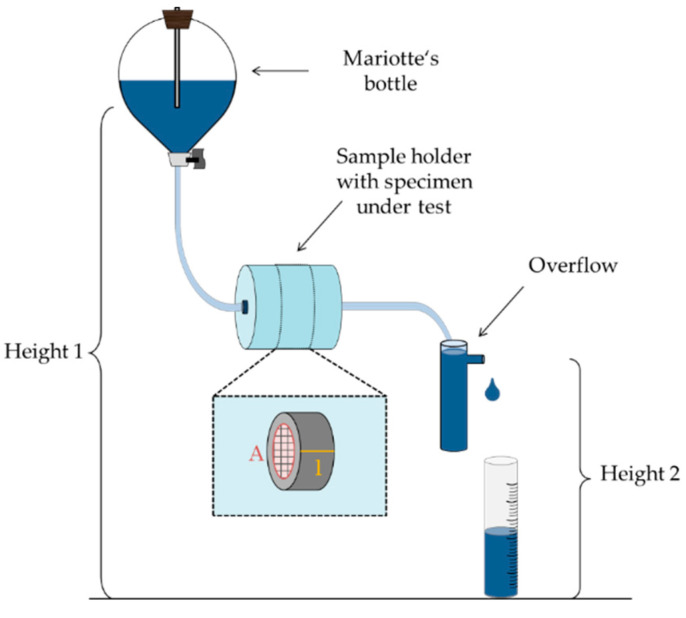
Experimental setup for permeability measurement ensuring defined hydrodynamic conditions.

**Figure 2 materials-14-05205-f002:**
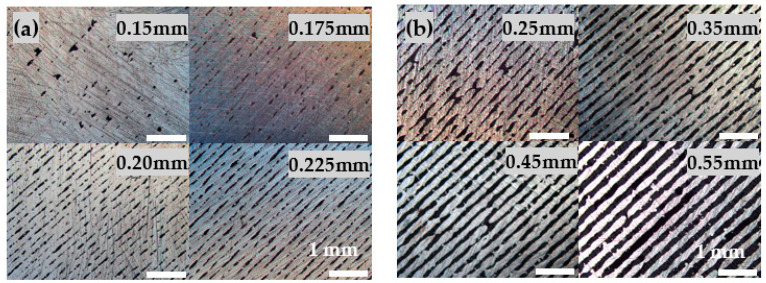
Formation of single walls in dependence of increasing hatch distance. (**a**) Insufficiently isolated hatch distances (0.15 mm to 0.225 mm) lead to not separated single walls. (**b**) As the hatch distances increases from 0.25 mm to 0.55 mm, single walls occur.

**Figure 3 materials-14-05205-f003:**
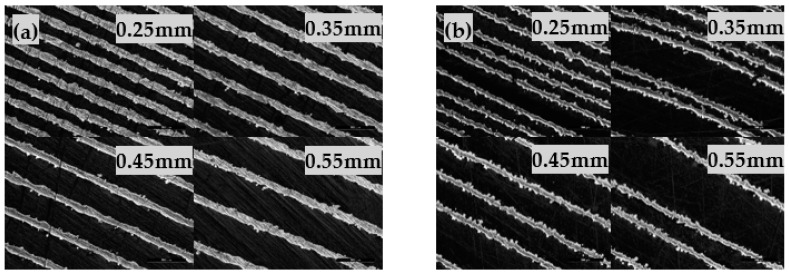
Single line walls. (**a**) Stable single walls due to suitable laser parameters (1390 m/s, 210 W). (**b**) Tilted walls and instable line formation due to insufficient energy density (2350 m/s, 180 W).

**Figure 4 materials-14-05205-f004:**
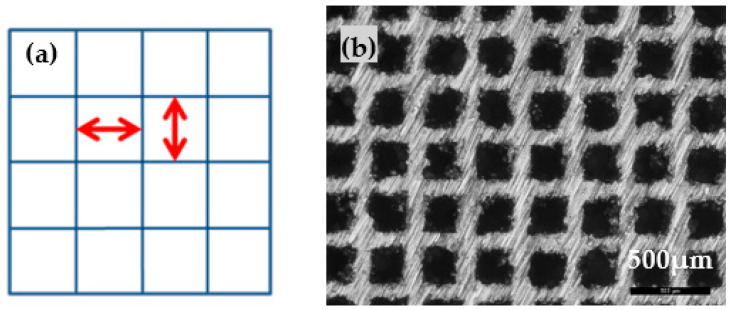
Square pattern. (**a**): Blue lines depict line scan of the laser, red arrows correspond to the applied hatch distance. (**b**): microscopic picture of an exemplary square pattern.

**Figure 5 materials-14-05205-f005:**
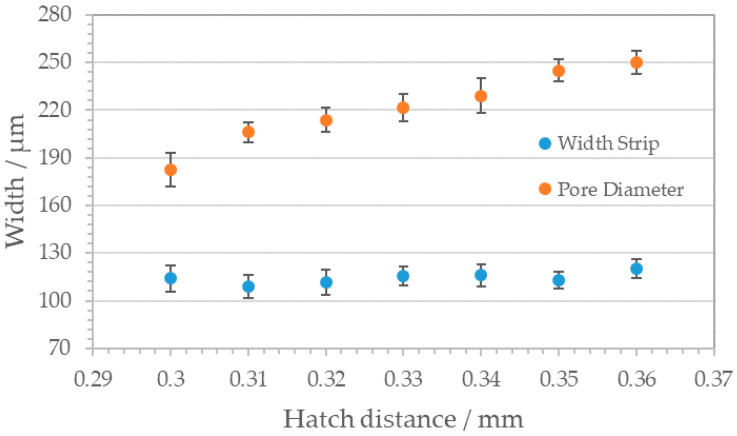
Correlation between hatch distance and resulting pore geometries for square pattern structure. Blue dots, width strip; red dots, pore diameter. With rising hatch distance, the pore diameters increase while the width strip remains constant.

**Figure 6 materials-14-05205-f006:**
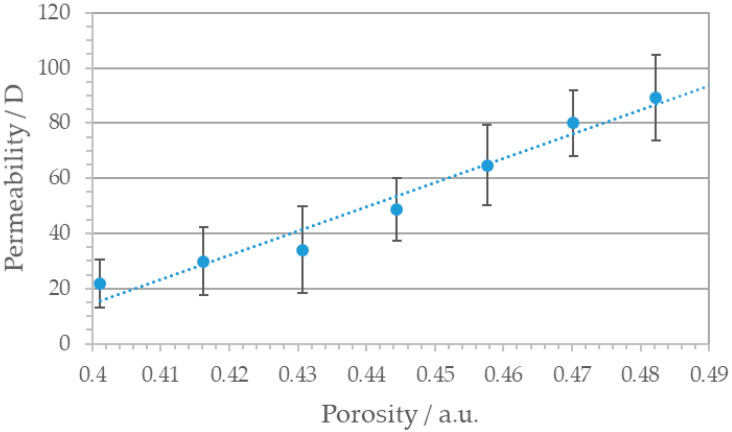
Correlation between permeability *K* and porosity of the square patterned SLM structures. With increasing porosity permeability rises from 20 D to nearly 100 D.

**Figure 7 materials-14-05205-f007:**
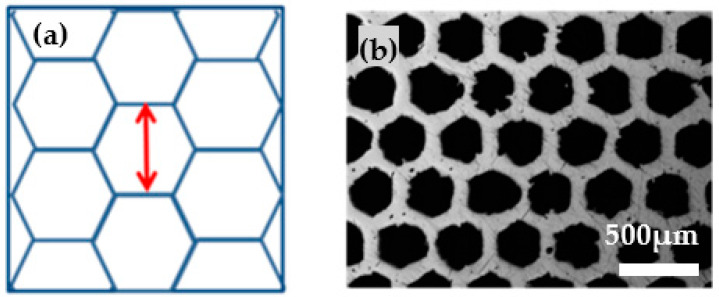
Honeycomb pattern. (**a**) Blue lines depict line scan of the laser, red arrows correspond to the applied hatch distance. (**b**) Microscopic picture of an exemplary honeycomb pattern.

**Figure 8 materials-14-05205-f008:**
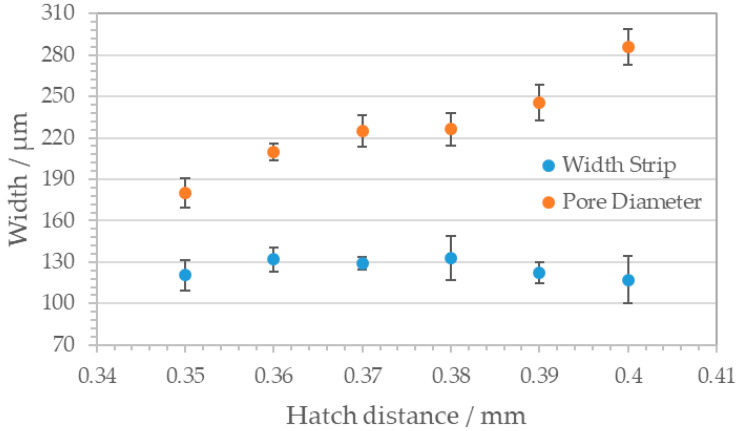
Correlation between hatch distance and pore geometry of the honeycomb pattern. Blue dots: width strip, red dots: pore diameter. With rising hatch distance, the pore diameters increase while the width strip remains constant.

**Figure 9 materials-14-05205-f009:**
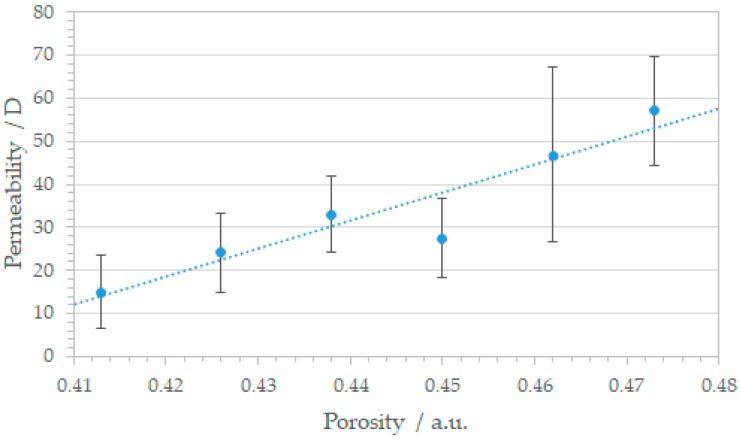
Correlation between permeability *K* and porosity of the honeycomb patterned SLM structures. With increasing porosity permeability rises from less than 20 D to nearly 60 D.

**Figure 10 materials-14-05205-f010:**
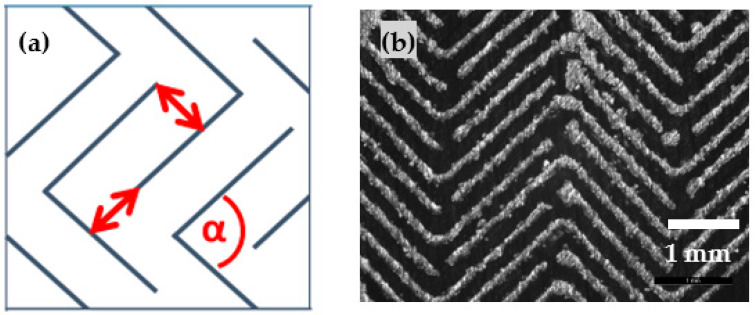
Anisotropic structure. (**a**) Schematic structure with laser line scan (blue lines), hatch distance (red arrow) and triangular opening angle α. (**b**) Microscopic image of the structure with an opening angle α = 90°.

**Figure 11 materials-14-05205-f011:**
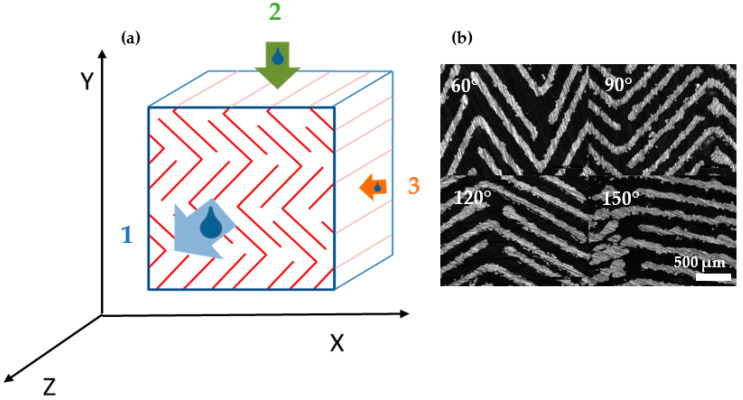
(**a**) Flow orientation in dependence of spatial structure orientation. Blue: direction 1 with unhindered flow independent of tilt angle α. Green: direction 2 with tunable flow ratio in dependence of tilt angle α—the more obtuse α, the more the flow is possible. Orange: direction 3 blocked flow direction. (**b**) Microscopic image of four angles α (clockwise starting upper left: 60°, 90°, 120°, 150°).

**Figure 12 materials-14-05205-f012:**
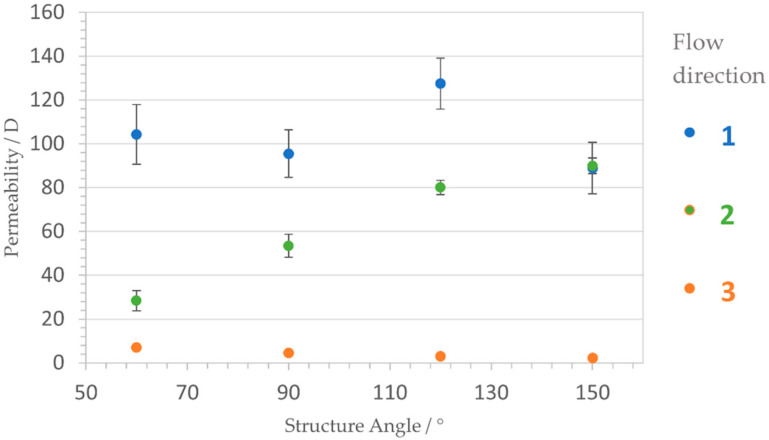
Correlation between permeability *K* and structure angle α, depicted in dependence of the dimensional orientation. Blue dots, unhindered orientation; green dots, angle dependent orientation; orange dots, blocked flow orientation.

**Figure 13 materials-14-05205-f013:**
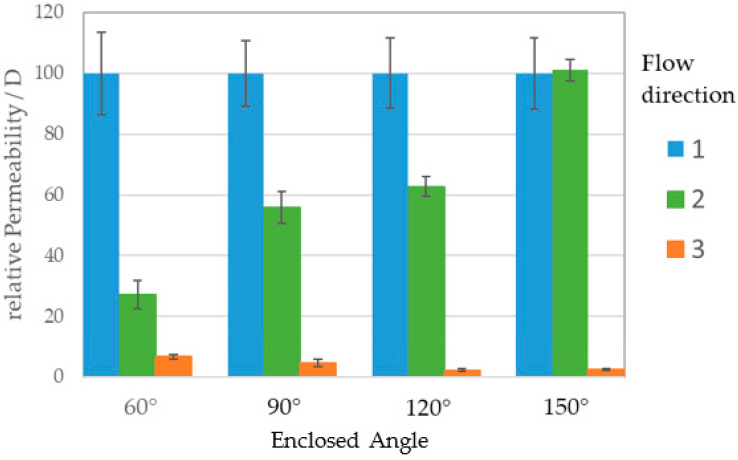
Flow ratio normalized to unhindered flow. With increasing tilt angle α the ratio of direction 2 (green) to 1 (blue) in rises from 0.25 up to 1.

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
