# Peer review of "Tunable, Anisotropic Permeability and Spatial Flow of SLM Manufactured Structures"

_materials, 2021, doi:10.3390/ma14185205_

Round 1

Reviewer 1 Report

Selective Laser Melting (SLM) is important for easy production of novel metal structures and many other applications.  A new approach was put forward by the authors to fabricate SLM structures with controlled permeability and other interesting characteristics. Solid experimental results on pattern geometry were satisfactorily achieved and well explained.
An experimental set up was also built to measure the permeability of the printed structures and a mathematical correlation between printing parameters was also explained.

The manuscript contains original research results with detailed information to readers.  English is very good and the manuscript is also well referenced. A slightly revised manuscript will attract many reads and most hopefully citations if it can be accepted by “Materials”.

Here is the reviewer´s minor-revision suggestion: in terms of laser technology, since optimized laser energy density must be adapted carefully to warrant wall stability etc. in the author´s experiment, is it possible for authors to add a New Figure about their selective laser melting setup including laser beam focusing optics etc. in order to facilitate the easy reading by readers from different research areas with different academic backgrounds?  Since laser beam quality M2 factors are very important for selective laser melting (if a TEM00 mode laser was used?), could the authors enrich their manuscript with more laser beam intensity and laser beam quality parameters for each pattern achieved.

Author Response

Thank you for your support, as well as the valuable comments and suggestions regarding our article „Tunable anisotropic permeability and spatial flow ratio control of defined single line Selective Laser Melting structures”. We are aware that these proposals to change the manuscript help to strengthen our paper.

We have modified the manuscript according to these comments and respond in detail as follows:

Comment:          Selective Laser Melting (SLM) is important for easy production of novel metal structures and many other applications.  A new approach was put forward by the authors to fabricate SLM structures with controlled permeability and other interesting characteristics. Solid experimental results on pattern geometry were satisfactorily achieved and well explained. An experimental set up was also built to measure the permeability of the printed structures and a mathematical correlation between printing parameters was also explained.

The manuscript contains original research results with detailed information to readers.  English is very good and the manuscript is also well referenced. A slightly revised manuscript will attract many reads and most hopefully citations if it can be accepted by “Materials”.

Here is the reviewer´s minor-revision suggestion: in terms of laser technology, since optimized laser energy density must be adapted carefully to warrant wall stability etc. in the author´s experiment, is it possible for authors to add a New Figure about their selective laser melting setup including laser beam focusing optics etc. in order to facilitate the easy reading by readers from different research areas with different academic backgrounds?  Since laser beam quality M2 factors are very important for selective laser melting (if a TEM00 mode laser was used?), could the authors enrich their manuscript with more laser beam intensity and laser beam quality parameters for each pattern achieved.

Answer:               We thank the reviewer for his kind comment and complemented the proposed laser characteristics as well as more detailed data in the experimental section.

As SLM appears to have advanced to a pretty common technique, we would prefer to refrain from adding another schematic illustration of the process and rather prefer to refer to a recent, nice publication of Nakano (Nakano, T. Selective Laser Melting. In Multi-dimensional Additive Manufacturing; Kirihara, S., Nakata, K., Eds.; Springer Singapore: Singapore, 2021; pp 3–26, ISBN 978-981-15-7909-7). As a good and relevant reference containing all further details for the described SLM process, we think it is indeed a valuable reference for interested readers.

Reviewer 2 Report

The paper submitted to Materials entitled “Tunable anisotropic permeability and spatial flow ratio control of defined single line Selective Laser Melting structures”, treats about the idea of building the structures with the use of additive manufacturing which will possess the anisotropic permeability.

The title reflects what is covered by the paper but my suggestion is to somehow simplify that. Maybe …”Tunable, anisotropic permeability in SLM manufactured structures” … something that is easier to catch . This is only a suggestion.

The paper is well planned and written in a proper way with the use of proper language and terminology.

No graphical abstract was provided – I encourage authors to do so.

The graphs and photos are in general of nice quality.

The scientific soundness of the paper is nice.

In general, I recommend publishing the paper with minor corrections and suggestions that I will provide below and authors may want to include them in the final version.

  • Please improve a bit the introductory part – for example showing some potential applications of the structures you design.
  • Please improve the description of the experimental part – with given description of the process, it would be difficult or impossible to replicate the experiment (fabricate samples). Maybe also provide files to supplementary data.
  • In figure 2 please more clearly indicate which sample is which
  • In figure 4 the hatch distance is still not clear – whether the distance is calculated from the middle of the wall or the surface, please improve the figure to be more clear
  • Im not sure how Materials treat that but in most journals the decimal separator is “dot”.
  • I don’t see error bars at figure 12 flow direction 3.
  • In figure 13 in one case it looks like flow ratio is >100%, also no error bars are given.

Author Response

Thank you for your support, as well as the valuable comments and suggestions regarding our article „Tunable anisotropic permeability and spatial flow ratio control of defined single line Selective Laser Melting structures”. We are aware that these proposals to change the manuscript help to strengthen our paper.

We have modified the manuscript according to these comments and respond in detail as follows:

Comment 1:       The paper submitted to Materials entitled “Tunable anisotropic permeability and spatial flow ratio control of defined single line Selective Laser Melting structures”, treats about the idea of building the structures with the use of additive manufacturing which will possess the anisotropic permeability.  

The title reflects what is covered by the paper but my suggestion is to somehow simplify that. Maybe …”Tunable, anisotropic permeability in SLM manufactured structures” … something that is easier to catch . This is only a suggestion.

Answer:               The reviewer raises an interesting concern and we understand that a short title may attract readers interest. Thus, we have changed the title to: “Tunable, anisotropic permeability and spatial flow of SLM manufactured structures”.

Comment 2:       The paper is well planned and written in a proper way with the use of proper language and terminology.

The graphs and photos are in general of nice quality.

The scientific soundness of the paper is nice.

No graphical abstract was provided – I encourage authors to do so.

Answer 2:           We thank the reviewer for his proposal and follow his suggestions. A graphical abstract was added.

In general, I recommend publishing the paper with minor corrections and suggestions that I will provide below and authors may want to include them in the final version.

Comment 3:       Please improve a bit the introductory part – for example showing some potential applications of the structures you design.

Answer 3:           We added a text passage to clarify the aim of our study. As the combination of regular SLM structures with resulting fluidic characteristics represents a new approach, applications using flow control are possible in terms of filtering, mixing of fluids or flushing without the necessity to remove the part.

Comment 4:       Please improve the description of the experimental part – with given description of the process, it would be difficult or impossible to replicate the experiment (fabricate samples). Maybe also provide files to supplementary data.

Answer 4:           We added the proposed information to the experimental part and also added a valuable reference, recently published, to guide the reader with deeper interest.

Comment 5:       In figure 2 please more clearly indicate which sample is which.

Answer 5:            We labelled the samples more clearly.

Comment 6:       In figure 4 the hatch distance is still not clear – whether the distance is calculated from the middle of the wall or the surface, please improve the figure to be more clear

Answer 6:           We added some text to better describe the mentioned parameters.

Comment 7:       Im not sure how Materials treat that but in most journals the decimal separator is “dot”.

Answer 7:            Thank you, that unwitting mistake was corrected for in all figures.

Comment 8:       I don’t see error bars at figure 12 flow direction 3.

Answer 8:           Flow direction 3 almost shows no passage of the fluid. Therefore, the time until the necessary volume is obtained is very long. Due to our mathematical analysis, the error bars are relatively small, consequently. We added the error bars in figure 13 to show the error bars more clearly.

Comment 7:       In figure 13 in one case it looks like flow ratio is >100%, also no error bars are given.

Answer 7:           We added the proposed error bars (see Answer 6). As direction 1 is set to 100% as a reference, a value above 100% only indicates a relative more fluid passage in direction 2.

Reviewer 3 Report

The authors investigated the tunable anisotropic permeability and spatial flow ratio control in various single-line designed SLM structures. The methodology, results and discussion, and the conclusion sections correlate well with each other. However, some improvements need to be made before this manuscript can be considered to be accepted, as follow:

1. Page 2, line 52: What is batch size one? Please provide some details/descriptions.

2. Page 2, lines 77-86: What do the authors actually want to say in this paragraph? At the moment, it is currently unclear. Perhaps the authors can improve this paragraph by re-wording or re-structuring the sentence.

3. Page 2, introduction section: The aim and objectives of this study should be mentioned in the introduction section, but it is currently absent. At present, the well-correlated methodology, results and discussion, and conclusions make less sense without clear aim and objectives.

4. Section 3 results and discussion: Please use letters such as a) and b) to describe all the figures, rather than left/right as it might be confusing. Please also amend the labels of the figures in text using letters accordingly.

Author Response

Thank you for your support, as well as the valuable comments and suggestions regarding our article „Tunable anisotropic permeability and spatial flow ratio control of defined single line Selective Laser Melting structures”. We are aware that these proposals to change the manuscript help to strengthen our paper.

We have modified the manuscript according to these comments and respond in detail as follows:

The authors investigated the tunable anisotropic permeability and spatial flow ratio control in various single-line designed SLM structures. The methodology, results and discussion, and the conclusion sections correlate well with each other. However, some improvements need to be made before this manuscript can be considered to be accepted, as follow:

Comment 1:       Page 2, line 52: What is batch size one? Please provide some details/descriptions.

Answer 1:           We modified the text passage to clarify our intended statement.

Comment 2:       Page 2, lines 77-86: What do the authors actually want to say in this paragraph? At the moment, it is currently unclear. Perhaps the authors can improve this paragraph by re-wording or re-structuring the sentence.

Answer 2:           We added a text passage to emphasize the aim of our study. The new aspect lies in the combination of defined structures, which often aim lightweight applications and porous SLM part, which are used as filtering objects. Our study deals with the combination of these two perspectives.

Comment 3:       Page 2, introduction section: The aim and objectives of this study should be mentioned in the introduction section, but it is currently absent. At present, the well-correlated methodology, results and discussion, and conclusions make less sense without clear aim and objectives.

Answer 3:           We added a text passage to emphasize the aim of our study. As the combination of regular SLM structures with resulting fluidic characteristics represents a new approach, applications using flow control are possible in terms of filtering, mixing of fluids or flushing without the necessity to remove the part.

Comment 4:       Section 3 results and discussion: Please use letters such as a) and b) to describe all the figures, rather than left/right as it might be confusing. Please also amend the labels of the figures in text using letters accordingly.

Answer 4:           We followed the reviewer suggestion and changed the labelling.